# Can Whole-Body Baseline CT Radiomics Add Information to the Prediction of Best Response, Progression-Free Survival, and Overall Survival of Stage IV Melanoma Patients Receiving First-Line Targeted Therapy: A Retrospective Register Study

**DOI:** 10.3390/diagnostics13203210

**Published:** 2023-10-14

**Authors:** Felix Peisen, Annika Gerken, Alessa Hering, Isabel Dahm, Konstantin Nikolaou, Sergios Gatidis, Thomas K. Eigentler, Teresa Amaral, Jan H. Moltz, Ahmed E. Othman

**Affiliations:** 1Department of Diagnostic and Interventional Radiology, Tuebingen University Hospital, Eberhard Karls University, Hoppe-Seyler-Straße 3, 72076 Tuebingen, Germany; isabel.dahm@med.uni-tuebingen.de (I.D.); konstantin.nikolaou@med.uni-tuebingen.de (K.N.); sergios.gatidis@med.uni-tuebingen.de (S.G.); ahmed.e.othman@googlemail.com (A.E.O.); 2Fraunhofer MEVIS, Max-von-Laue-Straße 2, 28359 Bremen, Germany; annika.gerken@mevis.fraunhofer.de (A.G.); alessa.hering@mevis.fraunhofer.de (A.H.); jan.moltz@mevis.fraunhofer.de (J.H.M.); 3Diagnostic Image Analysis Group, Radboud University Medical Center (Radboudumc), Geert Grooteplein Zuid 10, 6525 GA Nijmegen, The Netherlands; 4Image-Guided and Functionally Instructed Tumor Therapies (iFIT), The Cluster of Excellence (EXC 2180), 72076 Tuebingen, Germany; 5Max Planck Institute for Intelligent Systems, Max-Planck-Ring 4, 72076 Tuebingen, Germany; 6Center of Dermato-Oncology, Department of Dermatology, Tuebingen University Hospital, Eberhard Karls University, Liebermeisterstraße 25, 72076 Tuebingen, Germany; thomas.eigentler@charite.de (T.K.E.); teresa.amaral@med.uni-tuebingen.de (T.A.); 7Department of Dermatology, Venereology and Allergology, Charité—Universitätsmedizin Berlin, Corporate Member of Freie Universität Berlin and Humbolt-Universität zu Berlin, Luisenstraße 2, 10117 Berlin, Germany; 8Institute of Neuroradiology, Johannes Gutenberg University Hospital Mainz, Langenbeckstraße 1, 55131 Mainz, Germany

**Keywords:** melanoma, imaging biomarkers, prognostic biomarkers, artificial intelligence and machine learning, targeted therapy, biomarkers for targeted therapy

## Abstract

Background: The aim of this study was to investigate whether the combination of radiomics and clinical parameters in a machine-learning model offers additive information compared with the use of only clinical parameters in predicting the best response, progression-free survival after six months, as well as overall survival after six and twelve months in patients with stage IV malignant melanoma undergoing first-line targeted therapy. Methods: A baseline machine-learning model using clinical variables (demographic parameters and tumor markers) was compared with an extended model using clinical variables and radiomic features of the whole tumor burden, utilizing repeated five-fold cross-validation. Baseline CTs of 91 stage IV malignant melanoma patients, all treated in the same university hospital, were identified in the Central Malignant Melanoma Registry and all metastases were volumetrically segmented (*n* = 4727). Results: Compared with the baseline model, the extended radiomics model did not add significantly more information to the best-response prediction (AUC [95% CI] 0.548 (0.188, 0.808) vs. 0.487 (0.139, 0.743)), the prediction of PFS after six months (AUC [95% CI] 0.699 (0.436, 0.958) vs. 0.604 (0.373, 0.867)), or the overall survival prediction after six and twelve months (AUC [95% CI] 0.685 (0.188, 0.967) vs. 0.766 (0.433, 1.000) and AUC [95% CI] 0.554 (0.163, 0.781) vs. 0.616 (0.271, 1.000), respectively). Conclusions: The results showed no additional value of baseline whole-body CT radiomics for best-response prediction, progression-free survival prediction for six months, or six-month and twelve-month overall survival prediction for stage IV melanoma patients receiving first-line targeted therapy. These results need to be validated in a larger cohort.

## 1. Background

Owing to new drugs, the therapy of melanoma patients in an advanced stage has seen dramatic improvements in recent years. In addition to checkpoint inhibition (ipilimumab (CTLA-4), nivolumab/pembrolizumab (PD-1), and their combination) [1], blocking the RAF-RAS-MEK signaling pathway with BRAF inhibitors mono or combined BRAF and MEK inhibitors (targeted therapy) has been established. However, this is possible for less than half of patients only due to the specific mutation patterns required [2,3,4,5,6]. The application of targeted therapy agents has enhanced overall survival (OS) and progression-free survival (PFS) [7,8,9]. Unfortunately, 15–20% of tumors show primary resistance to this therapy, and patients can develop resistance to treatment [10]. Clinically validated predictive markers such as lactate dehydrogenase (LDH), S100B, and tumor burden exist to identify patients who do not benefit from targeted therapy [8,11]. On the other hand, experimental biomarkers based on radiomics are the focus of recent research to improve response and survival prediction. Radiomics describes the extraction of phenotypic features from medical image data using either handcrafted features or, more recently, abstract features from deep learning and the subsequent development of imaging biomarkers using machine- or deep-learning methods [12]. While several studies have investigated the use of radiomics and machine learning to support clinical decision-making in metastatic melanoma [13,14,15,16,17,18], there is still a demand for further investigation as limited data for patients treated with targeted therapy exist and no radiomic biomarker is yet established for routine clinical use [2,17].

The aim of the study was to investigate in a machine-learning prediction model if radiomics extracted from three-dimensional segmentation of all baseline metastases, combined with clinical parameters, offers Appendix A compared with clinical parameters alone. Using a volumetric whole-tumor-burden segmentation approach, we aimed to extract information also from small lesions and variables such as the whole-body tumor burden. The primary endpoints were best response, progression-free survival for six months, and overall survival for six and twelve months. 

## 2. Material and Methods

### 2.1. Workflow Framework

The workflow is depicted in Figure 1. In the baseline computed tomography (CT) images of the selected patient cohort, all metastatic lesion volumes were segmented by hand. A large set of radiomic features was extracted per lesion and aggregated across the lesions per patient. Two machine-learning (ML) models were trained and evaluated with a repeated five-fold cross-validation scheme. The baseline model used clinical data only; the extended model used clinical data and radiomic features. The models consisted of feature preprocessing, feature selection, training, and validation. 

### 2.2. Patient Selection

The institutional ethics board approved the study protocol. Patients diagnosed with metastatic malignant melanoma between January 2015 and December 2018 were retrospectively identified in the melanoma registry of the department of dermatology. First-line treatment was conducted at the department of dermatology, a tertiary referral center. Inclusion criteria were metastatic malignant melanoma stage IV, first-line treatment with targeted therapy (BRAF inhibitor monotherapy or in combination with a MEK inhibitor, according to current guidelines), existing baseline CT scans, documented demographic data, follow-up data, and clinical metadata. Exclusion criteria were missing baseline imaging with CT, previous treatment with targeted therapy, or no depictable metastasis on CT imaging. For an illustration of the inclusion and exclusion process see Figure 2. For a detailed description of the final cohort, refer to Table 1.

### 2.3. Imaging and Metastases Segmentation

The radiomic analysis was carried out on baseline CT data, which were downloaded from the local picture archiving and communication system (PACS). Inhouse staging CTs were performed on four CT scanners and one PET-CT scanner. The inhouse whole-body staging protocol was used with a scan field from the skull base to the middle of the femur with patients laid in a supine position, arms raised above the head. Scanning was performed during the portal-venous phase after administration of body-weight-adapted contrast medium through the cubital vein. For a detailed distribution of CT vendors and inhouse scan parameters, see Appendix A. To account for a more realistic sample and reduce sample bias, 26 baseline CTs of patients who had received their CT not inhouse at the department of diagnostic and interventional radiology, but at several external institutions, were included. Detailed information for contrast medium phase, tube current, and tube voltage are not available for these cases.

The image analysis was performed on transversely reformatted images of the portal-venous phase. Manual segmentation of all visible metastases was volumetrically performed by a radiologist (F.P., with 5 years’ experience in oncologic imaging) and verified by an experienced reader in cases of uncertainty using a four-eye principle (A.O., with 9 years’ experience in oncologic imaging). For this purpose, the anonymized data were imported into a custom-made segmentation software (SATORI, Fraunhofer MEVIS, Bremen, Germany). All clearly identifiable and measurable metastases, including 123 osseous metastases (see Table 1), were segmented to obtain the most reliable approximation of the total tumor burden. Cerebral metastases were not included in the segmentation, as the whole-body CT protocol could not replace a dedicated cranial computed tomography scan. However, the presence of brain metastases was included into the model as binary information.

### 2.4. Feature Extraction and Machine-Learning Model

For the radiomic feature extraction, aggregation, and development of a machine-learning model, we used a pipeline that we have described in depth in a previous publication [19]. Summing up, radiomic features were extracted for each segmented lesion using the reference standard software Pyradiomics (version 2.1.0) [12]. The lesion-wise features were aggregated per patient, to enable patient-level training and outcome prediction. In addition, the lesion count and total lesion volume were computed across all segmented lesions and per lesion type. In total, 5284 radiomic features were calculated per patient. Automatic feature selection using the fast correlation-based filter for feature selection (FCBF) method [20] was applied during training to select a small subset of features that have a high correlation with the outcome on training data and little correlation with other selected features. 

A random forest (RF) was chosen as the core ML model [21] using the implementation provided in the scikit-learn Python package (version 0.24.2, ExtraTreesClassifier) [22]. Two ML models (baseline and extended) were trained for the endpoints binarized best response (CR + PR vs. SD + PD after first cycle of targeted therapy according to RECIST 1.1 [23]), progression-free survival for six months, and overall survival after six months and twelve months, respectively. The baseline model was trained on the clinical data (features listed in Figure 3). The extended model was trained on all clinical features and a subset of the aggregated radiomic features that was automatically selected with the FCBF per fold. All ML models were trained in repeated 10 × 5-fold cross-validation (CV) with random assignment of patients to the folds to obtain a reliable estimate of the overall model performance. For more in-depth technical details regarding the radiomics and ML pipeline, refer to our previous publication [19].

### 2.5. Model Evaluation

For the model evaluation, the area under the curve (AUC) of the receiver operating characteristic (ROC) curve was selected. Bootstrapping with 1000 samples to estimate a 95% confidence interval (CI) for the mean AUC of the 10 × 5-fold CV of each model was used [24]. The difference in performance of the two models was regarded as statistically significant if the CIs of their mean AUCs did not overlap. Additionally, we divided the cohort into low- vs. high-risk patients: A patient was classified as high-risk when the twelve-month overall survival prediction of the respective ML model was negative. The lifelines Python package (version 0.26.0) was employed to compute a Kaplan–Meier estimator for the overall survival of both groups, with a log-rank test for the statistical comparison of both groups [25].

## 3. Results

### 3.1. Demographics, Best Response, Progression-Free Survival for Six Months, and Overall Survival after Six and Twelve Months

The study cohort had a median age of 56 years and was predominantly male (59%). Superficial spreading melanoma (55%) and nodular melanoma (34%) were the most frequent histological subtypes. Baseline LDH was elevated in 42%, and baseline S100B in 51% of the patients. Table 1 provides detailed patient characteristics.

### 3.2. Prediction of Best Response, PFS, and Overall Survival

#### 3.2.1. Model Evaluation

The radiomics model was numerically superior to the baseline model for the prediction of best response and PFS for six months (delta of the mean AUC for best response +0.064; delta for PFS +0.093) but not for the prediction of overall survival at six and twelve months (delta for overall survival after six months −0.094; delta for overall survival after twelve months −0.078); see Table 2. However, the difference was not significant, as there was an overlap of the confidence interval for all endpoints. Figure 4 shows the mean ROC curve for all endpoints and both models, where an overlap of the CI of the ROC curves can also be seen. Except for the baseline model for best response, all models indicated a prediction capacity (mean AUC > 0.5); however, only three models were significantly superior to random guessing (AUC 0.5 not included in CI). See Table 2.

#### 3.2.2. Risk Stratification for Overall Survival

Depending on the twelve-month overall survival prediction in the five-fold cross-validation, applicable patients were divided into a low-risk and a high-risk group using Kaplan–Meier estimators. Subsequent log-rank tests showed a significant distinction for the baseline model (*p* < 0.005) but not for the extended model (*p* = 0.43). See Figure 5.

## 4. Discussion

In this study, a previously developed machine learning radiomics pipeline for pre-treatment baseline CTs of patients undergoing immunotherapy [19] was applied to a cohort undergoing targeted therapy. For the immunotherapy cohort, we described the predictive capabilities of five ML models for three clinical endpoints (response at three months and overall survival at six and twelve months), but no statistically significant difference between the baseline and extended models. We also documented a trend (*p* = 0.06) for risk stratification into low-risk and high-risk groups for overall survival using the extended model.

In the present study’s targeted therapy cohort, the mean AUCs were in a comparable range; however, no significant predictive capabilities of the ML models could be shown, as indicated by the large confidence intervals. For best response and PFS, the results indicated a small increase in the AUCs when adding radiomic parameters. For overall survival prediction, adding radiomic parameters led to a drop in performance. As for the immunotherapy cohort, no significant differences between the baseline and extended models could be shown for the targeted therapy cohort. The baseline model was able to differentiate high-risk patients from low-risk patients for overall survival, correlating with the results published for immunotherapy cohorts [26,27]. With the extended model, no differentiation of low-risk and high-risk patients was possible.

The results show that the Appendix A gained by the extended ML model for best-response prediction and PFS prediction is limited, and for the endpoints six-month and twelve-month overall survival even detrimental to the performance, as demonstrated by the low AUCs and AUC differences. Missing information from the follow-up imaging and the connected change in the lesion parameters might explain this underperformance. Gassenmaier et al. [11] investigated levels of LDH and S100B during therapy in patients with metastatic melanoma at distinct time points. Dercle et al. [28] followed a mixed delta radiomics and clinical metadata approach and investigated a patient collective with advanced melanoma treated with immunotherapy. Both authors exemplarily showed the potential of the so-called delta approach. Our aim, however, was to make a statement at the earliest point of time possible, which is the baseline CT. There are promising baseline-focused publications available for melanoma patients treated with immunotherapy [26,27] and one publication investigating the role of baseline FDG PET/CT for response prediction in a mixed immunotherapy/targeted therapy melanoma cohort [29]. To our knowledge, no publications are available for the comparison of baseline clinical data and CT radiomics for a targeted therapy melanoma cohort.

Technical challenges like model overfitting [30], varying CT acquisition parameters between vendors and institutions, as well as differences in post-processing, might also explain the low performance of the extended model. However, one must keep in mind that boards in referral centers receive patients from different institutions with CTs that have been acquired from varying sources [31,32,33]. In our eyes, a more selective approach would have therefore not been beneficial for clinical transferability of the results. Nevertheless, the AUCs of all our extended ML models were not very high (<0.7), which indicates that there was no stable biomarker. This matches with previous results from a large cohort undergoing first-line immunotherapy [19].

The present study included a cohort of patients with prospective documentation of clinical metadata and long-term follow-up in a registry. The sample size was comparable to other radiomic studies investigating melanoma patients treated with targeted therapy [34] and immunotherapy [13,14,35,36]; however, the number of segmented lesions was larger in our cohort due to the whole-body segmentation approach [16]. The cohort consisted of a first-line treatment collective and exhibited the prescribed trend that patients with BRAF(+) are often younger and present with superficial spreading tumors or tumors with nodular histology, developing in anatomical regions without chronic sun damage [10]. The cohort showed a typical response distribution to targeted therapy. Median overall survival was in the reported range of a stage IV melanoma cohort treated with targeted therapy [7]. The random forest model was therefore developed and tested in a real-life cohort. By including different CT vendors and image data sources, selection bias was reduced. The whole-body segmentation approach was chosen to account for a potential loss of information from non-target lesions, which are usually not included in RECIST-based studies. All visible metastases in the baseline CTs were volumetrically segmented to gain the maximum information of the whole tumor load and to create as much information output as possible for the radiomics analysis as a benchmark for predictive performance. 

Even though the study used a large number of segmented metastases and a similar sample size compared to other radiomic studies, an extended collective with more independent samples would have been of value for the development of a radiomics ML model. The applied study protocol might have underestimated the true total tumor burden in some patients. Other modalities like FDG-PET/CT are more sensitive to the detection of certain types of metastases, such as osseous lesions. However, FDG-PET/CT is not the standard modality for staging in all melanoma patients yet, leading to smaller numbers of available subjects. We therefore decided to focus on widespread available CT and tried to compensate for the reduced sensitivity by consulting follow-up CTs and other staging modalities, if applicable, to identify as many lesions on baseline imaging as possible. Because there was a very large number of lesions and the outlining was carried out by hand, the volumetric segmentation was executed only once by just one reader (F.P.). Supervision by an experienced radiologist (A.O.) was given; however, a second reading and a second reader are missing. An assessment of inter- and intra-reader variability was therefore not possible. There is partially missing information about CT acquisition parameters because the study included external imaging sources to reduce selection bias, guarantee a realistic sample as requested by current radiological guidelines, and not further reduce the cohort size [37]. An external validation cohort is lacking. We aimed to compensate for that limitation through multiple cross-validations, an established method used in prior studies [38], and believe that an external validation cohort would not have delivered additional information, as the results are pointing toward a failure of the tested method. 

The results of the study mainly point toward a failure of the investigated method. However, through our study design and despite the negative results, we hope to address the recently discussed bias toward the publication of positive results in radiomic studies, lacking a comparison with a non-radiomic approach, as described by Kocak et al. [39].

## 5. Conclusions

The study applied a previously developed radiomics machine-learning pipeline to a stage IV melanoma patient cohort undergoing first-line targeted therapy. In the selected sample, it indicated no added value of radiomics in baseline CT imaging for best-response prediction, progression-free survival prediction for six months, or six-month and twelve-month overall survival prediction. The resulting AUCs remained low, even though all visible metastases in the whole body were volumetrically segmented. To improve the capacity of the current and other models, it is necessary to include larger cohorts with more independent samples and apply techniques combining different biomarkers and radiomic features. 

## Figures and Tables

**Figure 1 diagnostics-13-03210-f001:**
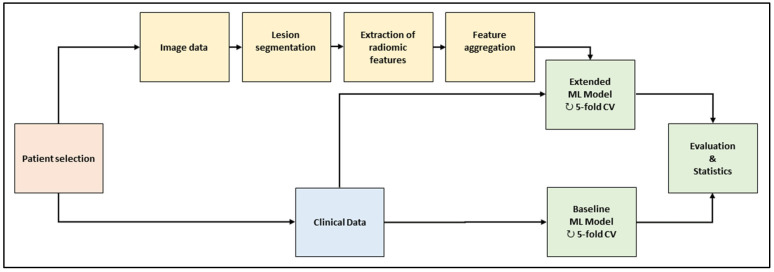
Framework of the machine-learning workflow. Abbreviations (ML: machine learning; CV: cross-validation).

**Figure 2 diagnostics-13-03210-f002:**
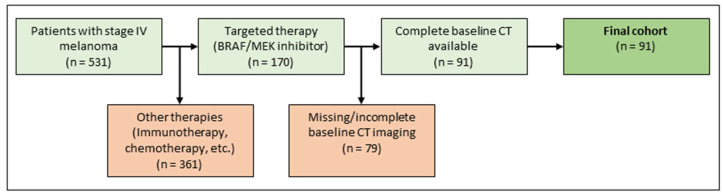
Patient selection.

**Figure 3 diagnostics-13-03210-f003:**
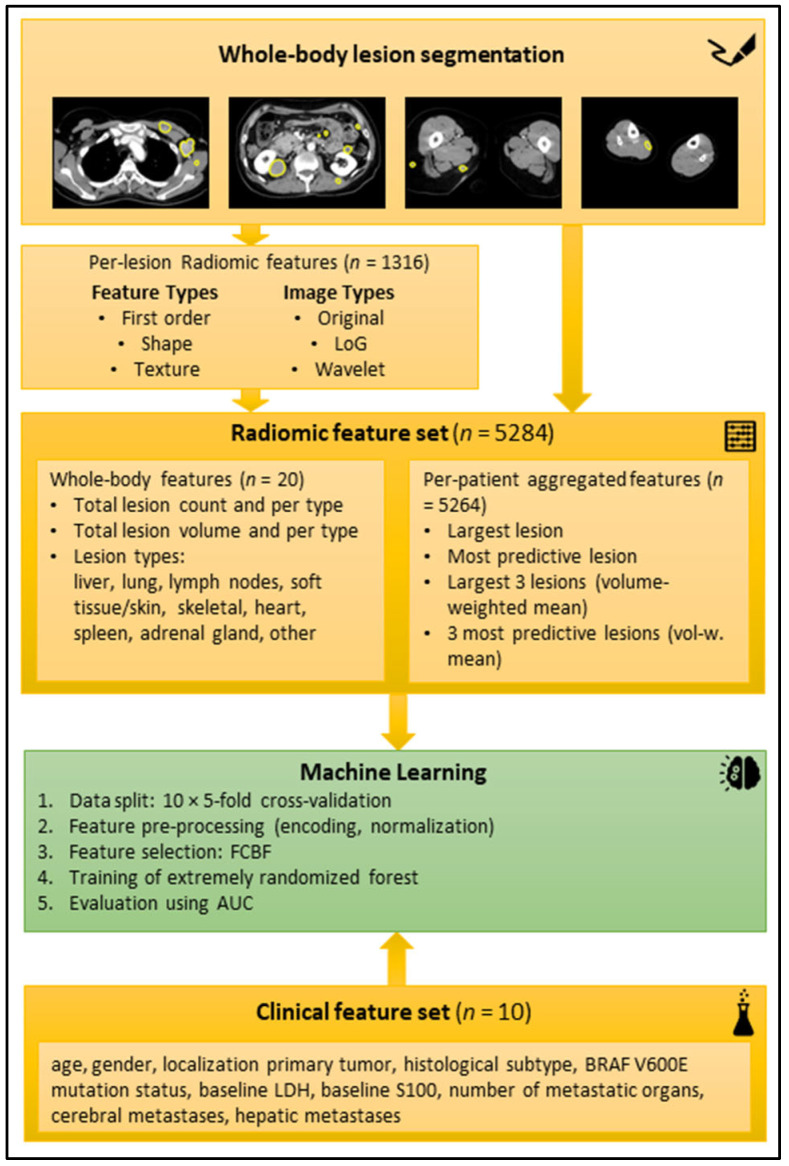
Schematic workflow of the radiomic analysis. Top yellow box: Examples of manual segmentations in axial reformatted CT slices in portal venous contrast medium phase. Second yellow box from top: radiomic features extracted per lesion. Third yellow box from top: Radiomic features aggregated per patient. Green box: machine-learning process. Bottom yellow box: Clinical feature set. Abbreviations: AUC (area under the curve), BRAF (v-Raf murine sarcoma viral oncogene homolog B1), FCBF (fast correlation-based filter), LDH (lactate dehydrogenase), LoG (Laplacian of Gaussian).

**Figure 4 diagnostics-13-03210-f004:**
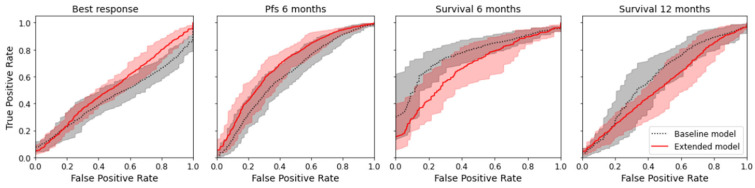
Mean ROC curves with 95% confidence interval for the true positive rate computed by bootstrapping the 10 × 5-fold CV.

**Figure 5 diagnostics-13-03210-f005:**
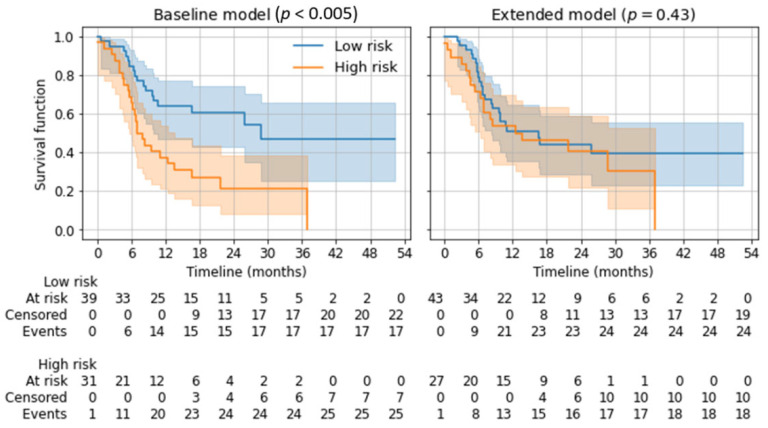
Kaplan–Meier estimators for low-risk and high-risk groups based on the predicted 12-month overall survival. *p*-values from log-rank tests are given for the separation of both risk groups.

**Table 1 diagnostics-13-03210-t001:** Patients’ characteristics.

Clinical Data		
Age (years) [median, (IQR)]		56 (21.5)
Gender (female) [*n*, %]		37 (41%)
Localization of primary tumor [*n*, %]	Head/neck	11 (12%)
	Torso	39 (43%)
	Upper extremity	5 (5%)
	Lower extremity	19 (21%)
	n/a	17 (19%)
Histological subtype [*n*, %]	SSM	50 (55%)
	NM	31 (34%)
	LMM	1 (1%)
	ALM	1 (1%)
	n/a	8 (9%)
BRAF V600E mutation status [*n*, %]	BRAF wildtype	0 (0%)
	BRAF mutation	91 (100%)
Baseline LDH [*n*, %]	Normal (<250 U/L)	33 (36%)
	Elevated (≥250 U/L)	42 (46%)
	n/a	16 (18%)
Baseline S100B [*n*, %]	Normal (<0.1 µg/L)	23 (25%)
	Elevated (≥0.1 µg/L)	51 (56%)
	n/a	17 (19%)
Number of metastatic organs [*n*, %]	1–3	70 (77%)
	>3	21 (23%)
Presence of cerebral metastases [*n*, %]		27 (30%)
Presence of hepatic metastases [*n*, %]		29 (32%)
Targeted therapy [*n*, %]	Dabrafenib	4 (4%)
	Dabrafenib + trametinib	43 (47%)
	Vemurafenib	1 (1%)
	Vemurafenib + cobimetinib	36 (40%)
	Encorafenib	1 (1%)
	Encorafenib + binimetinib	6 (7%)
Lesion counts [*n*]	All	4727
	Lung	2006
	Liver	753
	Soft tissue/skin	1155
	Lymph nodes	379
	Skeletal	123
	Spleen	99
	Other	212
Patient outcome		
Best response (RECIST 1.1) [*n*, %]	CR	11 (12%)
	PR	43 (47%)
	SD	15 (16%)
	PD	19 (21%)
	n/a	3 (3%)
Progression-free survival for 6 months [*n*, %]	Yes	36 (40%)
	No	44 (48%)
	n/a	11 (12%)
Overall survival after 6 months [*n*, %]	Yes	69 (76%)
	No	17 (19%)
	n/a	5 (5%)
Overall survival after 12 months [*n*, %]	Yes	37 (41%)
	No	34 (37%)
	n/a	20 (22%)
Overall survival (months) [median, (95%CI)]		26.2 (20.8–31.6)

Abbreviations: ALM, acral lentiginous melanoma; CR, complete response; IQR, interquartile range; LDH, lactate dehydrogenase; LMM, lentigo maligna melanoma; n/a, not available; NM, nodular melanoma;; PD, progressive disease; PR, partial response; RECIST, response evaluation criteria in solid tumors; SD, stable disease; SSM, superficial spreading melanoma.

**Table 2 diagnostics-13-03210-t002:** Number of cases with class distributions and mean AUC from a 10 × 5-fold CV and 95% confidence interval computed by bootstrapping the 10 × 5-fold CV. Class 0 best response = PD/SD; Class 1 best response = PR/CR. Class 0 PFS/OS = no; Class 1 PFS/OS = yes).

	Binary Endpoint			
	Best Response	PFS for 6 Months	OS at 6 Months	OS at 12 Months
n cases, (class 0, class 1)	88 (34, 54)	80 (44, 36)	86 (17, 69)	71 (34, 37)
Baseline model (clinical features), [AUC (95% CI)]	0.460 (0.362, 0.559)	0.586 (0.479, 0.693)	0.742 (0.631, 0.853)	0.600 (0.489, 0.709)
Extended model (clinical and radiomic features), [AUC (95% CI)]	0.524 (0.431, 0.615)	0.679 (0.579, 0.774)	0.648 (0.524, 0.769)	0.522 (0.424, 0.625)

Abbreviations: AUC, area under the curve; CI, confidence interval; CV, cross-validation; n, number; CR, complete response; OS, overall survival; PD, progressive disease; PR, partial response; SD, stable disease.

## Data Availability

Data may be made available after a reasonable and well justified request to Felix Peisen. Data cannot, however, be made freely available to the public, due to privacy regulations. Codes and materials used in this study may be made available for purposes of reproducing or extending the analysis, pending materials transfer agreements.

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
