# Peer review of "Can Whole-Body Baseline CT Radiomics Add Information to the Prediction of Best Response, Progression-Free Survival, and Overall Survival of Stage IV Melanoma Patients Receiving First-Line Targeted Therapy: A Retrospective Register Study"

_diagnostics, 2023, doi:10.3390/diagnostics13203210_

Round 1

Reviewer 1 Report

Dear editor,

I read the manuscript with great interest, which deals with a cutting-edge topic and currently  radiomics applications is considered a trend topic.

The manuscript is written very well, it is understandable and does not contain any particular linguistic errors (consider that I am not native English and therefore not able to highlight grammatical minor errors).

The methods and results are described accurately but the discussion is long-winded and also deals with the machine learning methodology, which however has been described in an article already published, so I would summarize these comments.

However, the authors specify the limitations of their study and the conclusions are consistent with the results.

Author Response

Response to comments of Reviewer #1:

Point 1: “… but the discussion is long-winded and also deals with the machine learning methodology, which however has been described in an article already published, so I would summarize these comments.”

  • We would like to thank the reviewer for that remark. We revised the discussion accordingly and have shortened the machine learning section. Moreover, we have tried to summarize the discussion in a more precise way, hopefully making it more readable and clearer.

We hope that the applied changes regain your agreement. In case that further clarifications or information should be needed, please do not hesitate to contact us any time.

Yours sincerely

Felix Peisen

Reviewer 2 Report

Dear colleagues,

I read your manuscript with great interest. I found the manuscript well written and the results well presented.

However I do have a major issue with your investigations: bone metastases

I couldn't find any comment regarding bone lesions throughout the manuscript. Given the large number of melanoma metastases it is hard to believe that none of the patients showed bone metastases prior to treatment (nor in the methods, in the graphs or discussion section). However, I believe the way to address the assessment or exclusion of bone lesions, e.g. non recist lesions is crucial since you chose an holistic/whole body approach. Talking about whole body, it is displayed that some of the patients had cerebral metastases, which obvisouly couldn't have been segmented on CT-scan. It think, it is important here to clearly report the type of lesions that couldn't have been segmented on CT-scan due to technique/resolution (I would assume bone, cerebral lesions) and whether they might have had an important on the total tumor burden or not.

Secondly, based on the above, I missed in the methods but above all the discussion section a comparison with FDG-PET/CT. One of the major advantage of FDG-PET/CT is exactly a better assessment of bone metastases. Depending on the amount of bone metastases you had prior to treatment in your cohort your data may have been biased in the first place since you didn't have the real tumor burden due to technique highlighting so the superiority of hybrid imaging over morphological modalities for this purpose.

I would be happy to read the revised version of your manuscript including full responses to the addressed issues.

Best of luck.

Author Response

Response to comments of Reviewer #2:

Point 1: “However I do have a major issue with your investigations: bone metastases.

I couldn't find any comment regarding bone lesions throughout the manuscript. Given the large number of melanoma metastases it is hard to believe that none of the patients showed bone metastases prior to treatment (nor in the methods, in the graphs or discussion section). However, I believe the way to address the assessment or exclusion of bone lesions, e.g. non recist lesions is crucial since you chose an holistic/whole body approach. Talking about whole body, it is displayed that some of the patients had cerebral metastases, which obvisouly couldn't have been segmented on CT-scan. It think, it is important here to clearly report the type of lesions that couldn't have been segmented on CT-scan due to technique/resolution (I would assume bone, cerebral lesions) and whether they might have had an important on the total tumor burden or not.”

  • We thank you for that comment. This is a point that needs clarification in the manuscript. We included 123 bone metastases into the analysis, as reported in the results section of the manuscript (please see table 1). The segmentation was not limited to lytic osseous metastases with soft tissue mass, as recommended by RECIST 1.1 (1, 2). Instead, all clearly identifiable and measurable metastases were segmented, to get as close to the true total tumor burden as possible. Bone metastases are often hardly detected on conventional CT. However, we had the advantage of follow up CTs to retrospectively identify osseous metastases on baseline CTs and, moreover, 13 patients underwent PET/CT staging (please compare table 1 in the supplement), improving the identification of osseous metastases on CT significantly. Regarding brain metastases, you are very right in supposing that they were not part of the segmentation process. Our institutional whole-body CT protocol applies a scan field from skull base to the middle of the femur and does not allow a sufficient segmentation of brain metastases as firstly the resolution is too low, and secondly only parts of the cerebrum are covered by the scan range. Dedicated cranial CTs would have been the option of choice for the segmentation of cerebral metastases, however, some patients initially underwent cerebral MRI, so there was an inconsistency regarding the imaging modality. We therefore did not include brain metastases into the segmentation process. However, the presence of brain metastases was included into the model as a binary information (please compare table 1). The types of lesions that were not segmented are now more clearly described in the methods section. We believe that the total tumor burden was not significantly affected by the exclusion of brain metastases and non-measurable bone metastases. More importantly, including these lesions would have affected the radiomics analysis in a negative way, as non-metastatic tissue might have been included in the segmentation process. For clarification, we added the following sentences to the methods section 2.3.: “All clearly identifiable and measurable metastases, including amongst others 123 osseous metastases (see table 1), were segmented, to get the most reliable approximation of the total tumor burden. Cerebral metastases were not included in the segmentation, as the whole-body CT protocol could not replace a dedicated cranial computed tomography scan. However, the presence of brain metastases was included into the model as a binary information.”

Point 2: “Secondly, based on the above, I missed in the methods but above all the discussion section a comparison with FDG-PET/CT. One of the major advantage of FDG-PET/CT is exactly a better assessment of bone metastases. Depending on the amount of bone metastases you had prior to treatment in your cohort your data may have been biased in the first place since you didn't have the real tumor burden due to technique highlighting so the superiority of hybrid imaging over morphological modalities for this purpose.”

  • Thank you for this interesting remark. It is very true that FDG-PET/CT indeed offers an improved detection of metastases, compared to conventional CT, especially when they are small or in the case of bone metastases (which is also true for MRI). The potential benefit of FDG-PET/CT staging is enormous. Several studies have investigated the role of PET/CT in the context of melanoma patients undergoing targeted- and immunotherapy (3-8). This is the reason why some melanoma patients at our institution undergo FDG-PET/CT staging, however, in our country FDG-PET/CT is not the standard modality for staging in all melanoma patients, yet. Therefore, the number of potential subjects for research purposes remains low. We therefore focused on CT parameters only and we believe that every CT study dealing with the topic of melanoma and radiomics struggles with this kind of bias, too (for example (9)). However, one must admit that osseous metastases are not totally invisible on conventional CT. Especially osteolytic metastases and osseous lesions with a high percentage of soft tissue mass, particularly when they exceed a certain size, are clearly depictable on conventional CT. Therefore, we do believe that through the staging with conventional CT, the tumor burden might by underestimated if small osseous metastases are present, but with respect to the overall tumor burden, this effect is most cases marginal. We now more precisely discuss and mention this potential bias and limitation: “The applied study protocol might have underestimated the true total tumor burden in some patients. Other modalities like FDG-PET/CT are more sensitive to the detection of certain types of metastases, such as osseous lesions. However, FDG-PET/CT is not the standard modality for staging in all melanoma patients yet, leading to smaller numbers of available subjects. We therefore decided to focus on widespread available CT and tried to compensate for the reduced sensitivity by consulting follow up CTs and other staging modalities, if applicable, to identify as many lesions on baseline imaging as possible.”

We hope that the applied changes regain your agreement. In case that further clarifications or information should be needed, please do not hesitate to contact us any time.

Yours sincerely

Felix Peisen

  1. Eisenhauer EA, Therasse P, Bogaerts J, Schwartz LH, Sargent D, Ford R, et al. New response evaluation criteria in solid tumours: revised RECIST guideline (version 1.1). Eur J Cancer. 2009;45(2):228-47.
  2. Costelloe CM, Chuang HH, Madewell JE, Ueno NT. Cancer Response Criteria and Bone Metastases: RECIST 1.1, MDA and PERCIST. J Cancer. 2010;1:80-92.
  3. Ayati N, Sadeghi R, Kiamanesh Z, Lee ST, Zakavi SR, Scott AM. The value of (18)F-FDG PET/CT for predicting or monitoring immunotherapy response in patients with metastatic melanoma: a systematic review and meta-analysis. Eur J Nucl Med Mol Imaging. 2021;48(2):428-48.
  4. Wong ANM, McArthur GA, Hofman MS, Hicks RJ. The Advantages and Challenges of Using FDG PET/CT for Response Assessment in Melanoma in the Era of Targeted Agents and Immunotherapy. Eur J Nucl Med Mol Imaging. 2017;44(Suppl 1):67-77.
  5. Seban RD, Nemer JS, Marabelle A, Yeh R, Deutsch E, Ammari S, et al. Prognostic and theranostic 18F-FDG PET biomarkers for anti-PD1 immunotherapy in metastatic melanoma: association with outcome and transcriptomics. Eur J Nucl Med Mol Imaging. 2019;46(11):2298-310.
  6. Nakamoto R, Zaba LC, Rosenberg J, Reddy SA, Nobashi TW, Davidzon G, et al. Prognostic value of volumetric PET parameters at early response evaluation in melanoma patients treated with immunotherapy. Eur J Nucl Med Mol Imaging. 2020;47(12):2787-95.
  7. Flaus A, Habouzit V, De Leiris N, Vuillez JP, Leccia MT, Perrot JL, et al. FDG PET biomarkers for prediction of survival in metastatic melanoma prior to anti-PD1 immunotherapy. Sci Rep. 2021;11(1):18795.
  8. Sachpekidis C, Kopp-Schneider A, Hassel JC, Dimitrakopoulou-Strauss A. Assessment of early metabolic progression in melanoma patients under immunotherapy: an (18)F-FDG PET/CT study. EJNMMI Res. 2021;11(1):89.
  9. Dercle L, Zhao B, Gonen M, Moskowitz CS, Firas A, Beylergil V, et al. Early Readout on Overall Survival of Patients With Melanoma Treated With Immunotherapy Using a Novel Imaging Analysis. JAMA Oncol. 2022;8(3):385-92.

Round 2

Reviewer 2 Report

Dear authors,

Thank your for the revised version of your manuscript.

I believe, the authors fully adressed my previous concerns.

I don't have further requests.

My congratulations,

Best regards.